# Desert Dust Contribution to PM$_{10}$ Loads in Styria (Southern Austria) and Impact on Exceedance of Limit Values from 2013–2018

**Marion Greilinger** [1,2,*]**, Johannes Zbiral** [2] **and Anne Kasper-Giebl** [2]

1   Department of Climate Research, Zentralanstalt für Meteorologie und Geodynamik (ZAMG),
    1190 Vienna, Austria
2   Institute for Chemical Technologies and Analytics, Vienna University of Technology, 1060 Vienna, Austria;
    johannes.zbiral@tuwien.ac.at (J.Z.); anneliese.kasper-giebl@tuwien.ac.at (A.K.-G.)
*   Correspondence: marion.greilinger@zamg.ac.at; Tel.: +43-1-36026-2232

**Abstract:** From a legislators point of view, the contribution of natural sources to PM$_{10}$ loads is relevant since their impact can be subtracted from the daily limit value of PM$_{10}$ as regulated in a working staff paper by the European Commission (EC), supporting the European Air Quality Directive (2008/50/EC). This work investigates its applicability for two stations in Austria over a time period of six years (2013 to 2018), as the occurrence of long-range transport of desert dust is observed on a regular base. Different stations and different statistical parameters were evaluated to determine the regional background load and subsequently the net dust load (NDL). Results reveal an adapted approach of the methodology described by the EC, using the +/− 15-day mean average of the PM$_{10}$ at the regional background station, together with threshold criteria to identify only desert dust affected days. The results of calculated NDLs were in good agreement with crustal loads determined on filter samples during two desert dust events in 2016. Thus, the application of the EC method for a region in Central Europe, which experiences a regular but less pronounced impact of desert dust than stations in the Mediterranean, is discussed.

**Keywords:** air quality; desert dust; mineral dust; PM exceedance; net dust load; background load; particulate matter

## 1. Introduction

Particulate matter (PM) concentrations are known to have severe impact not only on global climate and atmospheric chemistry ([1] and references therein) but also on human health [2,3]. Regarding the findings from these studies, the European Commission (EC) has established limit values for PM in the air quality directive 2008/50/EC. Therein, a daily limit for PM$_{10}$ of 50 µg/m$^3$ is stated, which may be exceeded on 35 days per year. Also for other PM fractions such as PM$_{2.5}$ limit values are given by the EC; however, within this study, only PM$_{10}$ is investigated.

Close to populated areas, exceedance of the limit values of PM$_{10}$ are caused mainly by anthropogenic sources such as energy production, traffic and industry, although natural sources can contribute as well [4]. The relative importance of the different sources depends strongly on regional and temporal scales, often the urban impact adds to a marked initial pollution load [5,6]. Especially in the Mediterranean region natural mineral dust sources are known as important contributors to PM$_{10}$ exceedance [7–9]. If exceedance of limit values can be attributed to a natural phenomenon such as particle intrusion from arid regions (e.g., the Sahara, or the Arabian or Lybian deserts), they can be discounted. The EC provides a commission working staff paper [10] describing this procedure

for Spain. In the underlying study of Escudero et al. [11] a methodology for quantifying the daily African $PM_{10}$ load during dust outbreaks for Southern Europe is proposed, using Spain as study area. They determined the regional background load via applying a monthly moving 30th percentile to the $PM_{10}$ load at a regional background site, excluding the days affected by African dust transport and validated the methodology via crustal loads determined by chemical speciation of $PM_{10}$ filters. Within the working staff paper of the EC it is clearly stated that the use of this approach has not been validated for other countries and that no certainty exists on its accuracy for application. A recent study of Barnaba et al. [7] investigates the applicability of the EC methodology for Italy, describing limitations and drawbacks in the identification of desert dust days as well as in the quantification of the regional background load. Specific solutions for Italy are proposed and introduced. In their adapted methodology, they use each site as reference for its own background load, reduced the time window over which the background load is determined and introduced an automatic, model-based method to identify desert dust affected days.

It is well known that desert dust outbreaks regularly occur in Austria [12–14], although their impact is less pronounced than in the Mediterranean region. Their influence on air quality has not been investigated so far and a discussion of the EC methodology for the subtraction of desert dust contributions is completely missing.

This study presents the first validity check of the applicability of the EC methodology for Austria. This is of high relevance as both facts, exceedances of short-term limit values of $PM_{10}$ and the influence of desert dust occur in Austria. The critical application of a methodology developed for the Mediterranean region to a country within Central Europe is of special interest for continuative use. We evaluate whether long-range transport of desert dust can be quantified via the EC methodology and whether the respective subtractions would influence the exceedance of the short-term limit value for $PM_{10}$, taking the urban-traffic sampling site Graz Don Bosco as main example. We compared three different stations for their suitability to be regarded as the regional background station, required for applying the EC methodology. Furthermore, different statistical parameters for the computation of the net dust load that can then be discounted from the exceeding daily $PM_{10}$ load of the respective station were tested, as suggested by the EC. The calculated amounts of the net dust loads of two desert dust events in 2016 were compared with measured mineral dust loads based on chemical analyses of the respective filters to verify the results.

The evaluation comprises a six-year data basis from January 2013 to December 2018 and is focused on days with $PM_{10}$ concentrations > 50 μg/m$^3$, exceeding the daily limit value. The possible impact of desert dust to the overall load of $PM_{10}$ would be interesting, but goes beyond the scope of this study.

## 2. Materials and Methods

### 2.1. Study Area

Overall $PM_{10}$ concentrations in Austria show a declining trend [15]. Nevertheless $PM_{10}$ concentrations in Styria, especially at sites within its capital city Graz, regularly exceed the short-term limit value [16]. Therefore, the region around Graz was selected as study area. Graz is located in Styria, in the southern part of Austria, southeast of the main ridge of the Alps (Figure 1) and is the second largest city in Austria, with about 300.000 inhabitants. Exceedances of $PM_{10}$ short-term limit values are mainly due to the specific orographic situation and the combination of local pollution sources, the regional transport of particulate matter and the possible input of long-range transport.

The climate of Graz can be classified as continental but its orography plays a key role in the atmospheric dynamics, and hence, in the air quality of the city. The location in a valley basin facilitates a shielding effect of the Alps, and therefore, a lack of wind especially during winter, preventing vertical mixing. This makes the occurrence of inversions and fog more likely [17].'Due to the increased occurrence of inversion conditions, PM concentrations are supposed to be increased regularly leading to an exceedance of limit values. To highlight the importance of emission reduction measures the

applicability and impact of the EC methodology for an urban-traffic station Graz Don Bosco (DB) and an urban-background station Graz Süd (GS) was investigated on a six-year data basis from January 2013 to December 2018. Three stations (Masenberg, MB; Bockberg, BB; Lustbühel, LB) were taken into account as potential regional background station for the calculation of the net dust load following the EC methodology. Filter samples of a rural (Gratwein, GW) and a suburban site (Graz Ost GO), sampled during two desert dust episodes in February and April 2016, together with filters of DB for the event in February only, were used to validate the calculated net dust loads.

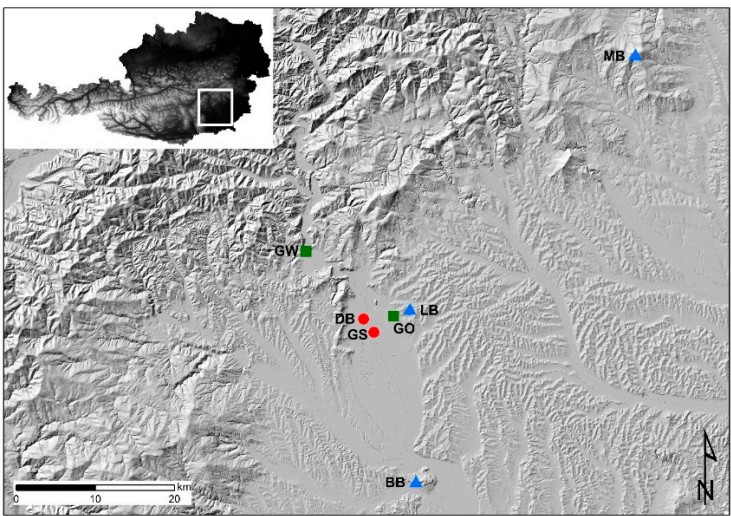

**Figure 1.** Location of the air quality monitoring stations selected for this study as listed in Table 1. Stations of investigation (Graz Don Bosco, DB, and Graz Süd, GS) are marked by red dots. Potential background stations, mandatory for the determination of the net dust load as proposed by the European Commission, are marked with blue triangles. Stations were filter samples for validation purposes were taken are marked via green squares.

## 2.2. $PM_{10}$ Measurements and Sampling

$PM_{10}$ concentrations at DB as well as at GS were gravimetrically measured according to the European standard reference method EN12341:2014 using a Digitel high volume sampler DHA80. $PM_{10}$ measurements at MB, BB and LB were derived using a beta attenuation mass monitor MetOne BAM 1020 (EN 16450:2017) The equivalence of this method to the reference method has to be proven and is shown in the annual reports of the environmental agency [18] and the provincial government of Styria [19].

At the stations GO and GW $PM_{10}$ samples of the two mineral dust episodes in February and April 2016 were collected on quartz fiber filters (Pallflex Tissuquartz) using a Digitel high volume sampler DHA80. Again, sampling was carried out according to EN12341:2014. The same filter material was used in DB to investigate the event in February. Sample changes, maintenance of the stations and gravimetric analysis were performed by the specialist department of the provincial government of Styria within the framework of the ambient air quality monitoring. Table 1 summarizes the location, measured parameters as well as the measurement and sampling devices.

**Table 1.** Monitoring stations used in the present study. See locations in Figure 1.

| Station | LON [°] | LAT [°] | Elevation [m a.s.l.] | Device |
|---|---|---|---|---|
| Graz Don Bosco (DB) | 15.41643 | 47.05702 | 358 | Digitel HVS DHA80 |
| Graz Süd (GS) | 15.43306 | 47.04167 | 345 | Digitel HVS DHA80 |
| Graz Lustbühel (LB) | 15.49369 | 47.06700 | 473 | MetOne BAM 1020 |
| Bockberg (BB) | 15.49583 | 46.87139 | 449 | MetOne BAM 1020 |
| Masenberg (MB) | 15.88222 | 47.34806 | 1180 | MetOne BAM 1020 |
| Graz Ost (GO) | 15.46638 | 47.05944 | 366 | Digitel HVS DHA80 |
| Gratwein (GW) | 15.32361 | 47.13555 | 382 | Digitel HVS DHA80 |

*2.3. Chemical Filter Analysis*

XRF analysis was used to determine the crustal load (CL) on the $PM_{10}$ quartz fiber filters sampled during the two dust events in February and April 2016. Measurements were performed using a Panalytical Axios Advanced wavelength dispersive X-ray fluorescence spectrometer with a Rhodium target X-ray tube. The tube was set at 50 kV with a current of 50 mA and a 20 mm aperture for exposure and an exposure time of 20 seconds per channel was used. Filter samples were put into a holder with a central opening of 27 mm in diameter. A set of eight filter holders with one filter blank and seven samples were put into the instrument and were automatically transferred to the analytical chamber one by one. Following the procedure described by Peng et al [20] a certified standard soil (San Joaquin soil from National Institutes of Standards, USA, NIST SRM 2709) was used for calibration. Concentrations of Mg, Ca, Fe and Al were determined and the respective crustal components such as $SiO_2$, $Al_2O_3$, $Fe_2O_3$, $CaCO_3$ and $MgCO_3$ were stoichiometrically calculated. The content of $SiO_2$ was indirectly determined from the content of Al using the relation $SiO_2 = 2*Al_2O_3$ [21]. Potassium was not included in the calculation of the CL, as it is well known that wood combustion used for residential heating is a dominant contributor to particulate matter concentrations within the region [22,23]. The CL was then computed by adding the concentrations of the major crustal components ($SiO_2$, $Al_2O_3$, $Fe_2O_3$, $CaCO_3$ and $MgCO_3$) similar to the approach of Escudero et al. [11] proposed in the EC guideline.

## 3. Results and Discussion

*3.1. $PM_{10}$ Levels at the Stations Graz Don Bosco (DB) and Graz Süd (GS)*

Daily $PM_{10}$ levels recorded at the stations DB (urban-traffic) and GS (urban-background) exceed the 50 µg/m$^3$ on 242 and 196 days, respectively, within the six-year period from January 2013 to December 2018, thereby exceeding the 35 day/year limit in six years and one year, respectively. The time series are plotted in Figure 2 showing that maximum $PM_{10}$ values, and hence, exceedance of limit values mostly occur during wintertime. This pattern of the seasonal variation is observed at both stations and is characteristic for stations in this region. The increase of PM concentrations during winter can on one hand be attributed to an enhanced influence of inorganic secondary aerosols due to lower air temperature and increased humidity and also to a stronger influence of wood combustion used for residential heating, as well as regional transport of PM [22,24]. Furthermore, meteorological conditions with wintertime inversions favor an enrichment of PM concentrations in the boundary layer. During wintertime inversion conditions are very likely in the area of Graz due to the orographic situation of the region, allowing air pollutants to concentrate over several days until the inversion is cleared out and concentrations drop [17].

On top of these seasonal variations the influence of long-range transported desert dust may occur. As an example, an intensive desert dust episode in April 2016 is marked by a red star in Figure 2. The identification, intensity and duration of this event, covering large parts of Austria, was already thoroughly described by Baumann-Stanzer et al. [12]. Still a number of less pronounced events took place. Such events, identified via the use of WRF-Chem model forecasts [12], FLEXTRA

back-trajectories [25] as well as the dust ensemble forecasts and forecast comparisons provided by the WMO sand and dust storm warning advisory and assessment system (https://sds-was.aemet.es/forecast-products/dust-forecasts), are marked in orange in Figure 2. Additionally optical aerosol properties measured at the high mountain global GAW station Hoher Sonnblick in the Austrian Alps are used for the identification of desert dust (DD) occurrence [14]. The aim of this study was the quantification of DD contributions for days with $PM_{10}$ exceedances only. An additional influence of desert dust might have occurred during periods of lower $PM_{10}$ concentrations as well.

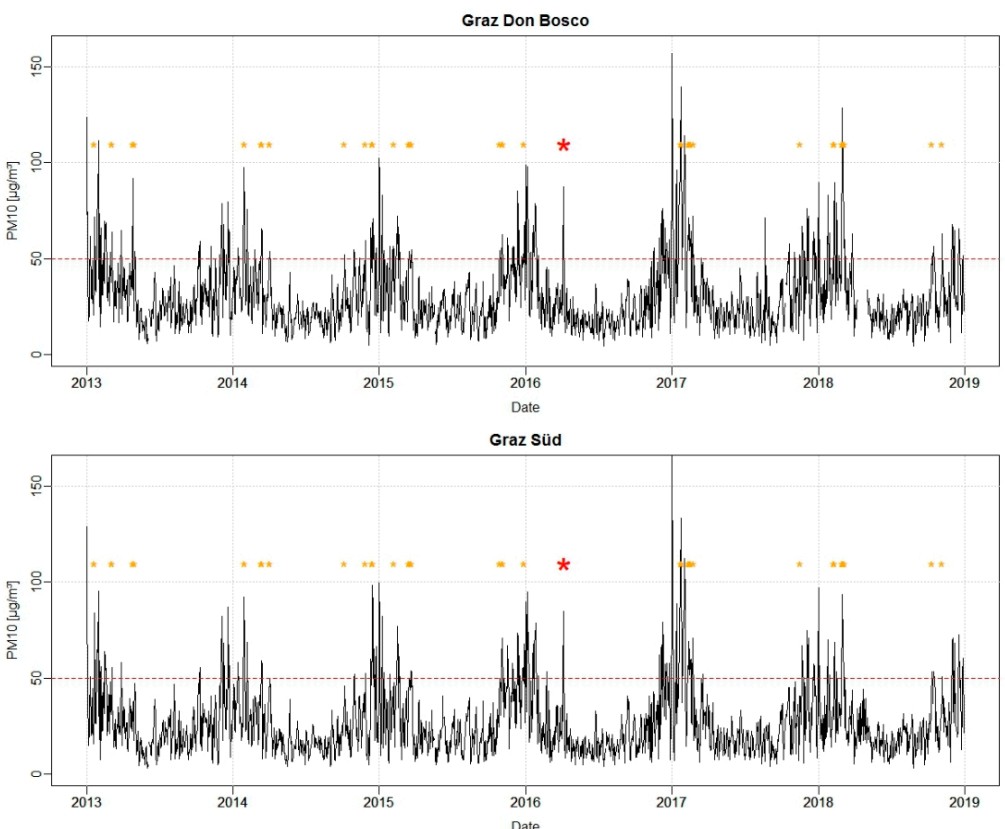

**Figure 2.** Particulate matter ($PM_{10}$) time series of the two stations under investigation. The red line marks the 50 µg/m$^3$ limit value set by the European Commission. Orange stars mark desert dust events on days where the $PM_{10}$ concentration was > 50 µg/m$^3$. The red star marks an intense Saharan dust event in April 2016.

### 3.2. Determination of Background Loads According to the EC Methodology

The EC methodology [10] for the evaluation of desert dust (DD) contributions is based on two steps. Firstly, dates affected by DD transport have to be identified and secondly, the net dust load (NDL) on the daily $PM_{10}$ record has to be quantified. The identification of DD events is based on several supporting information like trajectory analysis, model forecasts and/or satellite data, which have to be screened and interpreted. A number of models, back-trajectory calculations or data bases for satellite data are suggested by the EC methodology, but later evaluations included other sources as well [7,11] showing the technical and organizational change in that field. In case the various information portals give diverse results for the identification of a DD event further evaluations and comparisons are needed. The second step, the quantification phase, requires a continuous $PM_{10}$ monitoring at a single selected regional background site representative for the region under investigation, providing the reference or background load (BGL). The BGL represent a moving 30-day average value (mean, median or 40th percentile as suggested by the EC) of the 15 days before and 15 days after the investigated day, where days influenced by DD are excluded. The NDL at the regional background site, representing the

contribution of DD to PM$_{10}$, is then estimated as difference between the measured daily PM$_{10}$ load and the calculated BGL:

$$NDL_{RB} = PM10_{RB} - BGL_{RB} \qquad (1)$$

Within a first approach to estimate the influence of DD for Austria we do not exclude days with DD influence for calculating the NDL. This is a difference to the EC methodology and can be regarded as a more conservative approach. Doing so, we want to consider two main facts. Even if DD is a driver for BGLs at the respective sites and respective days in Austria, other sources might contribute markedly as well and an exclusion of days with influence of DD would eliminate these effects as well. Secondly, modeling tools used for the characterization of DD reveal some uncertainty, which influences the determination of days which could be rejected. This uncertainty might be more pronounced for Austria than for regions closer to the source. The uncertainty is supposed to increase with longer transport time because atmospheric dispersion of air pollutants is a very complex process with very high uncertainties in chemical transport models. Thus, we did not put this elimination step to the beginning of our evaluation procedure. As a matter of fact we will get a potential overestimation of the BGL at the regional background station and hence an underestimation of the according NDL for the respective day. Thus, a potentially lower value of a DD contribution is subtracted compared to reality. Some discussion of the systematical error introduced by this approach will be given later.

Regarding the quantification phase, we compared the 30-day mean, median and 40th percentile for the computation of the BGL at three potential regional background stations (compare Figure 3). The calculated BGL based on the mean is generally higher than the BGL based on the median or the 40th percentile, regarding all three stations. As may be expected from Equation (1), the estimated NDL decreases as the percentile for the computation of the BGL increases. Again, we decided to stick to a more conservative approach and use the 30-day mean for further computation of the BGL. Using the median or the 40th percentile would reduce the average BGL by 4–11% and 13–21%, respectively, depending on the station used (MB, BB or LB). The BGL calculated based on the 30-day mean varies between 10–50 μg/m$^3$ for BB and LB and between 4–21 μg/m$^3$ for MB and is reduced to 5–45μg/m$^3$ (BB and LB) and 3–19 μg/m$^3$ (MB) when the 40th percentile criterion is applied.

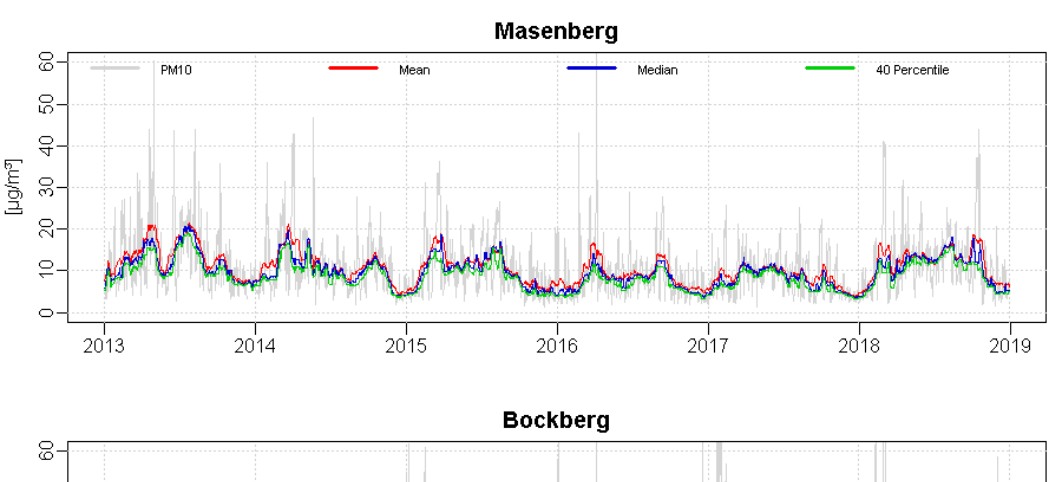

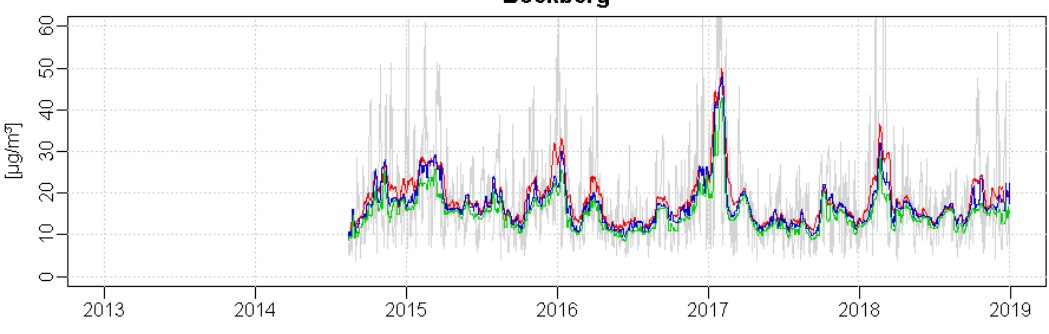

**Figure 3.** *Cont.*

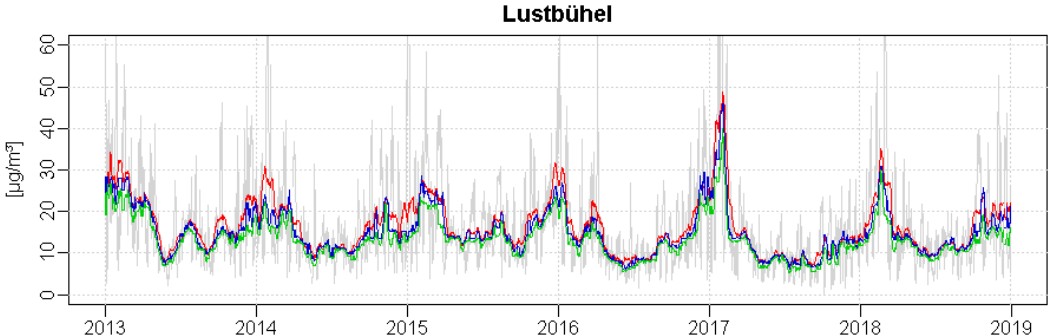

**Figure 3.** Background load (BGL) of three potential background stations using either the mean (red), the median (blue) or the 40th percentile (green) of the 30-day period for calculation.

### 3.3. Selection of a Suitable Regional Background Station

According to Equation (1) the NDLs of the respective days, calculated as the difference between $PM_{10}$ and BGL, will vary around zero. Negative values represent days where the BGL is larger than the central $PM_{10}$ value and vice versa.

In Figures 3 and 4 it can be seen that the time series of MB feature different characteristics than the one of BB and LB. $PM_{10}$ values of BB and LB tend to cover a wider range compared to MB and show a seasonality similar to DB and GS with increased values especially during winter. Correspondingly, BGLs of BB and LB (Figure 3) show this seasonality as well and NDLs (Figure 4) yield a much larger scatter during the wintertime, than during summer. Negative NDL outliers at BB and LB were found to occur mostly during winter and could be traced back to days at the end of an inversion weather situation where the $PM_{10}$ concentration suddenly drops due to an air mass exchange. Days with inversion weather conditions, meaning an inversed vertical temperature profile with lower values at lower altitudes compared to higher ones, were identified using the daily mean temperature at the TAWES stations at the airport of Graz (337 m a.s.l.) and on top of Schöckel (1445 m a.s.l.). If the daily mean temperature at the airport of Graz was lower than the one at Schöckel the day was classified as inversion day. Inversion days occur all years whereas their number varies from year to year. Thus, for single years and during short time measurement campaigns BB and LB might serve as suitable background stations for Graz, but not over a longer period of several years. Positive NDL outliers at BB and LB are of course identical with $PM_{10}$ outliers and occur mostly during winter, again reflecting the inversion weather situation where $PM_{10}$ concentrations are enhanced. Due to this seasonal dependence, BB and LB are ineligible as regional background stations to compute the NDL. The calculated NDL at these stations cannot be interpreted as such due to the substantial influence of inversion weather conditions. It rather reflects a "pollution load" pointing to a variety of influencing factors from anthropogenic and natural sources.

NDLs of MB are similar between winter and summer and, as a matter of fact, for the whole data set. Negative NDL outliers were found to be associated to days with precipitation rather than inversion weather situations shortly before or after. During precipitation the $PM_{10}$ concentration suddenly drops and becomes lower than during the previous and subsequent days representing the BGL. Positive outliers of the NDL are again identical with outliers of $PM_{10}$, but were quite equally distributed between the different seasons. The single outlier for the BGL was observed on July 25, 2013 related to a Saharan dust episode a few days later. Due to averaging over 30 days, also other days are influenced by this Saharan dust episode. Their BGL values were found to be only slightly smaller than the upper whisker and are therefore not highlighted.

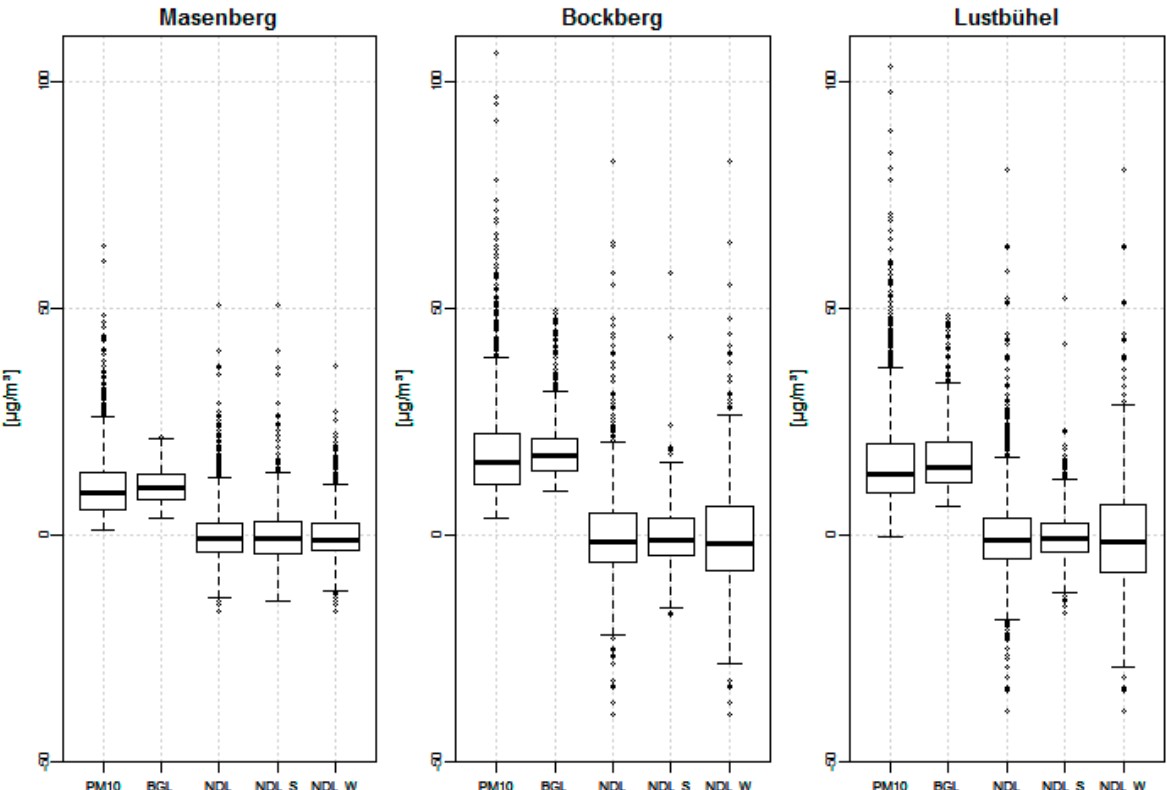

**Figure 4.** Boxplot of the daily mean $PM_{10}$, the background loads (BGL) calculated based on the mean as well as the net dust loads (NDL) for the whole years as well as for the summer (NDL_S) and winter period (NDL_W) of the three potential background stations.

Results show that MB represents a suitable background station because it is largely independent of inversion weather situations and hence seasonality. Based on an urban climate analysis study of Graz [17] it can be assumed that the representativeness of the background condition observed on MB is applicable for the stations in Graz. Thus, we investigate the contribution of desert dust on $PM_{10}$ measurements using MB to compute the BGL based on the mean of a 30-day period without extracting desert dust affected periods. This approach represents a modified methodology compared to the one proposed by the EC (40th percentile, exclusion of DD days) which was established for Spain, but the demand for its validation for other countries is clearly highlighted.

Due to its definition (see Equation (1)) the NDL is computed via a subtraction of the BGL from the daily $PM_{10}$ value and is likely to vary around zero. BGL values (representing a 30-day average $PM_{10}$ load) might be higher than the $PM_{10}$ load of a single day due to precipitation or increased vertical mixing after a period of stable atmospheric conditions on that special day, leading to negative NDL values. These negative values are just a consequence of the day to day variability and are, therefore, set to zero for further evaluation. Hence, the NDL of MB can be interpreted as a reduction potential in $\mu g/m^3$ due to the influence of desert dust lowering $PM_{10}$ levels observed at other stations in the investigated region. Table 2 gives an overview of the main statistical parameters describing the NDL before and after negative values were set to zero.

**Table 2.** NDL in µg/m$^3$ of the Masenberg station, most suitable to act as regional background station, before and after negative values were set to zero.

|  | **Before Setting Negative NDLs to Zero** | **After Setting Negative NDLs to Zero** |
|---|---|---|
| Minimum | −16.82 | 0 |
| 25th percentile | −3.79 | 0 |
| Median | −0.91 | 0 |
| 75th percentile | 2.79 | 2.79 |
| Maximum | 50.60 | 50.60 |

*3.4. Application of the Modified Methodology*

Table 3 shows the number of days exceeding the 50 µg/m$^3$ daily PM$_{10}$ limit at the two investigated stations DB and GS as well as the number of days exceeding this limit after the subtraction of the NDL$_{MB}$. Throughout the six-year period 242 days exceed the daily limit value at the urban-traffic station DB, which comes up to five out of six years exceeding the 35 day/year limit. At the urban-background station GS 196 days exceed the daily limit value, whereas only in 2017 the 35 day/year limit was exceeded due to a longer lasting pollution episode in January and February.

Subtracting the NDL$_{MB}$ without any further evaluation of the presence of DD, 40 days of the 242 days at DB fall below the 50 µg/m$^3$ limit, reducing the years which exceed the 35 day/year limit to three years (2013, 2017 and 2018) in contrast to the five years given before. For GS 35 days of 196 days fall below the 50 µg/m$^3$ limit, not influencing the exceedance of the 35 day/year limit. After an identification of DD days based on model result, back trajectories and optical aerosol measurements as described earlier, results reveal that of the 40 and 35 days at DB and GS, only 20 (DB) and 15 (GS) days show an influence of DD (compare Table 3, columns DD days). Obviously only for those days a subtraction of the NDL$_{MB}$ is allowed and the solely determination of NDL$_{MB}$ is not sufficient.

The selection of DD days could be included before subtracting the NDL$_{MB}$, corresponding to the EC methodology. This would identify 51 out of 242 days at DB and 37 out of the 196 days at GS with a possible influence of DD. Subtractions of NDL$_{MB}$ for these special days would again give the number of reductions listed in Table 3 (5th column for DB and 9th column for GS).

As already discussed, the identification of DD days using models together with back trajectories is a very subjective process. Barnaba et al [7] developed an automated and user-independent process for the identification of DD days, still using model calculations. As such, a user-independent process was not yet established for Austria. Therefore, we evaluate a method for the identification of DD days based on the time series of the PM$_{10}$ measurements at the background site only.

**Table 3.** Number of days with daily PM$_{10}$ > 50 µg/m$^3$ at the urban-traffic station Graz Don Bosco (DB) and the urban-background station Graz Süd (GS), number of days which are exceeding or falling below the 50 µg/m$^3$ limit value after the subtraction of the NDL of the regional background station MB (NDL$_{MB}$) without any further evaluation as well as with the identification of desert dust (DD) days via model calculations. Amount of days exceeding the limit of 35 days/year with PM$_{10}$ concentrations > 50 µg/m$^3$ are marked in bold.

| Year | PM$_{10}$ DB > 50 µg/m$^3$ | PM$_{10DB}$-NDL$_{MB}$ > 50 µg/m$^3$ | PM$_{10}$ DB-NDL$_{MB}$ ≤ 50 µg/m$^3$ | DD Days DB | PM$_{10}$ GS > 50 µg/m$^3$ | PM$_{10}$ GS-NDL$_{MB}$ > 50 µg/m$^3$ | PM$_{10}$ GS-NDL$_{MB}$ ≤ 50 µg/m | DD Days GS |
|---|---|---|---|---|---|---|---|---|
| 2013 | **44** | **36** | 8 | 4 | 31 | 25 | 6 | 1 |
| 2014 | 27 | 16 | 11 | 7 | 23 | 18 | 5 | 3 |
| 2015 | **39** | 30 | 9 | 5 | 35 | 29 | 6 | 3 |
| 2016 | **39** | 35 | 4 | 2 | 34 | 29 | 5 | 3 |
| 2017 | **54** | **49** | 5 | 1 | **43** | **37** | 6 | 1 |
| 2018 | **39** | **36** | 3 | 1 | 30 | 23 | 7 | 4 |
| Sum | 242 | 202 | 40 | 20 | 196 | 161 | 35 | 15 |

Considering only days with a possible contribution of DD similar features of NDL$_{MB}$ and BGL$_{MB}$ become visible, which define additional criteria to avoid undue reductions even when only NDL$_{MB}$

values are considered. The resulting thresholds ($BGL_{MB}$: 12 µg/m$^3$ and $NDL_{MB}$: 10 µg/m$^3$) are based on a descriptive approach, but give a better understanding of the several factors influencing $NDL_{MB}$.

$BGL_{MB}$ concentrations, determined as a 30-day average, should not fall below 12 µg/m$^3$, a value close to the annual average concentration of $PM_{10}$ determined at Masenberg. We want to point out that monthly averaged $PM_{10}$ values of the whole period of 6 years vary between 6–8 µg/m$^3$ for November, December and January and between 10–14 µg/m$^3$ for all other months. Setting a threshold for $BGL_{MB}$ at 12 µg/m$^3$ will preclude the determination of a DD event valid for reduction, during the winter month and also during most of the year 2017. Still a lower threshold value would lead to the identification of events not associated with the influence of DD, but by other sources or general differences between the background site at 1180 m asl and the urban sites in Graz. Furthermore, the calculated $NDL_{MB}$ should be at least 10 µg/m$^3$ and thus account for a substantial part of the $PM_{10}$ concentration, at least at the background site. This can be regarded as a precondition as the method should identify DD events which influence the air quality in Austria markedly. Ohterwise, low concentration values more affected by random variations could account for undue reductions of $PM_{10}$ levels. This is especially important during days when daily limit values at polluted sites are exceeded only slightly. The identified $NDL_{MB}$ threshold is close to the 95th percentile of 10.9 µg/m$^3$.

These thresholds were applied to the 242 days exceeding the 50 µg/m$^3$ daily $PM_{10}$ limit within the six-year period at DB only, since this is the more critical station regarding the 35 day/year limit. Now the amount of days still exceeding the limit value is very well comparable as if DD days were identified using model and trajectory analysis. An influence on the exceedance of the 35 day/year limit remains scarce (compare column 3 and 5 in Table 4) and occurs only in one year (2015) when the adapted methodology using the thresholds led to no reduction below the 35 day/year limit, whereas it would fall below if the identified DD days were used.

The general agreement and disagreement with the evaluation of identified DD days is given as contingency table (see columns 6 to 10 in Table 4), while evaluations of the single days are given in the supplement (Table S1).

**Table 4.** Column 1: Amount of days of $PM_{10}$ exceedance at DB, Column 2: Identified desert dust days (iDD), amount of DD days leading to a reduction below 50 µg/m$^3$ shown in brackets, Column 3: Remaining days exceeding the daily limit after deduction of the $NDL_{MB}$ on identified DD days, Column 4: Estimated desert dust days (eDD) due to the threshold criteria, amount of estimated DD days leading to a reduction below 50 µg/m$^3$ shown in brackets. Column 5: Remaining days exceeding the daily limit after deduction of the $NDL_{MB}$ on estimated DD days Column 6–10: Contingency table (TP = true positives, TN = true negatives, FP = false positives, FN = false negatives, NAs = no $PM_{10}$ data from Masenberg available).

| | 1 | 2 | 3 | 4 | 5 | 6 | 7 | 8 | 9 | 10 |
|---|---|---|---|---|---|---|---|---|---|---|
| Year | $PM_{10\,DB}$ > 50µg/m$^3$ | Identified DD Days (iDD) | $PM_{10\,DB}$ > 50µg/m$^3$ iDD Days | Estimated DD Days (eDD) | $PM_{10\,DB}$ > 50µg/m$^3$ eDD Days | TP | TN | FP | FN | NAs |
| 2013 | 44 | 8 (4) | 40 | 6 (5) | 39 | 3 | 32 | 3 | 4 | 2 |
| 2014 | 27 | 10 (7) | 20 | 4 (4) | 23 | 4 | 17 | 0 | 6 | 0 |
| 2015 | 39 | 10 (5) | 34 | 3 (3) | 36 | 3 | 28 | 0 | 7 | 1 |
| 2016 | 39 | 2 (2) | 37 | 2 (2) | 37 | 2 | 37 | 0 | 0 | 0 |
| 2017 | 54 | 11 (1) | 53 | 0 (0) | 54 | 0 | 43 | 0 | 10 | 1 |
| 2018 | 39 | 10 (1) | 38 | 6 (1) | 38 | 4 | 27 | 2 | 6 | 0 |
| Sum | 242 | 51 (20) | 222 | 21 (15) | 227 | 16 (6.6%) | 184 (76.0%) | 5 (2.1%) | 33 (13.6%) | 4 (1.7%) |

In total 16 days (6.6%) were identified as true positive (TP) DD days, whereas 184 days (76.0%) were identified as true negatives (TN). This means, that 82% of the days exceeding the daily limit were correctly classified in DD and non-DD days using the thresholds. Regarding the remaining days, five days (2%) were identified as false positive (FP) DD days, while 34 days (13.6%) were found to be false negatives. For 20 of these days some uncertainty of the classification of DD days based on model and

trajectory analysis remains, i.e., the visualizations showed slightly different results or the influence can be expected to be very small. The other 14 days show a clear influence of DD. Still all 34 days need further investigations, as this classification as false negatives could easily be an effect of the systematic error introduced by our calculation of the BGL. Adjusting the calculation procedure for every DD event could overcome this limitation, but this adjustment could also make the identification of DD days slightly more subjective.

For 15 out of these 34 days the $NDL_{MB}$ was below 2 µg/m$^3$ while $PM_{10}$ concentrations at DB ranged from 51 to 139 µg/m$^3$ with a mean value of 64 µg/m$^3$, indicating on one hand that DD is not the main influence on $PM_{10}$ concentrations at this site, on the other hand that an increase of $NDL_{MB}$ via a reduction of the BGL would allow to reach a reduction below the daily limit value in single cases. A repeated evaluation of the model and trajectory results revealed that for these 15 days the DD identification could easily be biased by subjectivity since results from the models and trajectories do not show a clear picture. An additional four days show an $NDL_{MB}$ below 4 µg/m$^3$ while $PM_{10}$ concentrations at DB range from 56 to 92 µg/m$^3$—again pointing to both statements. These days could be clearly identified as DD days and thus point to a limitation of the threshold method. Thus, the evaluation of single events remains of great importance, but should be supported by a chemical analysis of the crustal loads as given below exemplarily for two cases.

### 3.5. Validation of the $NDL_{MB}$ Based on Two Case Studies in 2016

In order to validate the $NDL_{MB}$ chemical analyses of filter samples collected at three sites (Graz Don Bosco, DB; Graz Ost, GO; Gratwein, GW; compare Figure 1) were used to determine the crustal loads. This analysis was performed for three days (23.02.2016, 05.04.2016 and 06.04.2016) featuring desert dust intrusion in was thoroughly described earlier.

For the strong transport event of desert dust in April, measured CLs match the calculated $NDL_{MB}$ very well, indicating that the $NDL_{MB}$ is mainly composed of mineral matter. CLs were found to account for more than 70% of the $NDL_{MB}$ with values ranging from 41.2 µg/m$^3$ at GO to 39.0 µg/m$^3$ at GW for the 5th of April 2016, compared to the $NDL_{MB}$ of 50.6 µg/m$^3$ and from 31.5 µg/m$^3$ at GO to 25.3 µg/m$^3$ at GW for the 06th of April 2016, compared to the $NDL_{MB}$ of 35.3 µg/m$^3$.

For the dust event in February, measured CLs are more than 50% lower (17.7 µg/m$^3$ at DB, 16.5 µg/m$^3$ at GO and 10.9 µg/m$^3$ at GW) than the calculated $NDL_{MB}$ (37.1 µg/m$^3$) pointing to additional contributions of other particulate matter sources. This result underlines the need of the threshold criteria. Using those the $NDL_{MB}$ is not allowed to be considered for a reduction, because the BGL is too low. Based on WRF-Chem model forecasts is can also be expected that the SD intrusion during this event was rather weak and other, maybe anthropogenic sources, dominate the NDL.

## 4. Summary and Conclusions

The study investigates the applicability of the methodology suggested by the European Commission (EC) to assess the contribution of desert dust (DD), to $PM_{10}$ loads in the region of Graz in Southern Austria over a time period of six years from January 2013 to December 2018.

The station Masenberg (MB) could be identified as a suitable regional background station and the $BGL_{MB}$ was calculated without an exclusion of DD affected days. Furthermore thresholds for the calculated $BGL_{MB}$ and the $NDL_{MB}$ of 12 µg/m$^3$ and 10 µg/m$^3$ were used to identify whether the $NDL_{MB}$ is really defined by DD and may be subtracted. We want to point out that this procedure is a more conservative approach than given in the EC guideline, representing the highest computable BGL, and thus the lowest NDL.

A detailed investigation of all days exceeding the daily limit $PM_{10}$ concentration showed that the risk of undue reductions is rather small. Within 16% of days which were not correctly classified, concentrations were not reduced for 33 days (false negatives), while an undue reduction happened only during five days (false positives). The evaluation of these false negative days reveals the limitations of

the method. Both calculation of NDLs and their restriction via thresholds and interpretation of models and trajectories might be biased.

It is obvious that using the proposed approach the $BGL_{MB}$ will be overestimated as soon as the DD influence holds longer than one day, leading to a corresponding underestimation of the $NDL_{MB}$. Still, the calculated $NDL_{MB}$ is higher than the crustal loads chemically measured on quartz fiber filters sampled during two DD events in 2016.

Overall the influence of the subtraction of natural sources (i.e., desert dust) from $PM_{10}$ concentrations at two sites in Graz yielded a number of reductions; however, on an annual basis, changes of the amount of days exceeding the daily limit were found to be rather small.

Within this study, we investigated stations in the region of Graz, representing a hot spot regarding $PM_{10}$ concentrations in Austria. Due to the limitation to days exceeding the daily limit, the impact on mean annual concentrations of the proposed approach was not assessed. Additionally, the applicability of the proposed approach to other stations needs further investigation and remains of great importance, but should be supported by a chemical analysis of the crustal loads.

**Supplementary Materials:** The following are available online at http://www.mdpi.com/2076-3417/9/11/2265/s1, Table S1: Detailed investigation of the days where the subtraction of the NDL MB reduced the $PM_{10}$ load at the urban-traffic station Don Bosco (DB) below the limit value of 50 μg/m$^3$.

**Author Contributions:** Conceptualization of the study was done by M.G. and A.K.-G. XRF filter measurements were made available by J.Z., who also helped to interpret the data. M.G. processed all the data. The manuscript was drafted by M.G. together with A.K.-G., with all co-authors commenting on the results and the manuscript content.

**Funding:** This research was part of the project DUSTFALL funded by the Austrian Research Promotion Agency (FFG), grant number 848858. The APC was funded by TU Wien University Library through its Open Access Funding Program.

**Acknowledgments:** The authors wish to acknowledge the provincial government of Styria, especially A. Schopper, for the air quality monitoring data from their operational network. Thanks go to Anton Neureiter for drawing the maps in ArcGIS, Erich Neuwirth for the XRF calibration and analysis of the filters, Claudia Flandorfer and Marcus Hirtl for the operational WRF-Chem model forecasts, Kathrin Baumann-Stanzer and Paul Skomorowski for the operational computation of the back-trajectories as well as Gerhard Schauer for the operational computation of the Saharan dust index at Sonnblick. The APC was funded by TU Wien University Library through its Open Access Funding Program.

**Conflicts of Interest:** The authors declare no conflict of interest. The funders had no role in the design of the study, in the collection, analyses, or interpretation of data; in the writing of the manuscript, or in the decision to publish the results.

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
