# Peer review of "Desert Dust Contribution to PM10 Loads in Styria (Southern Austria) and Impact on Exceedance of Limit Values from 2013–2018"

_applsci, doi:10.3390/app9112265_

Reviewer 1 Report

In the manuscript ID applsci-499476 entitled “Desert dust contribution to PM 10 loads in Styria (Southern Austria) and impact on exceedance of limit values from 2013-2018)” authors evaluate log-range transport of desert dust identified and quantified via EC in Styria.  The manuscript is well written and a lot of results are presented and discussed: I suggest minor revision before publication, listed below:

1.       In general, across the text: pay attention to subscripts (for example in PM10)

2.       Line 51-61: I suggest moving this paragraph in the “Study Area” section

3.       Line 86: Cn authors describe these kind of sites (Gratwein, GW and Graz Ost GO): are they traffic/background/rural sites?

4.       Line 102: can authors better describe the gravimetric procedure used?

5.       Line 178-179: Can authors better explain this statement?

Finally, I strongly suggest to reports in the manuscript the innovativeness of the study, motivations and scientific contribution of the study. 

Author Response

The authors thank the reviewer for his appreciated recommendations to our manuscript. We tried to follow them throughout the manuscript and hope for a positive decision on the submission status.

Reviewer 2 Report

The paper deals with a methodology to assess the contribution of desert dust to PM10 values in southern Austria. However, it is difficult to follow the arguments, since the implications of the main points are not fully explored and are not always supported by the experiments/data presented.

Furthermore, the language is difficult to follow at times and the structure of the arguments does not highlight the relevant points and conclusions. Therefore, the work is not of archival quality in its present form. This work may be publishable upon addressing the several major concerns raised below as well as many minor concerns regarding language and nomenclature. In short, the paper could be much better.

1. Introduction:

What is the motivation for the study?

Line 55:"reach the short term limit" - it seems that according to e.g. line 60 the short term limits are exceeded? What is true?

Materials and Methods:

2.1 Study Area:

What were the selection criteria for selecting the measurement stations?

PM10 measurement and sampling: How were the PM10 measurements done? Was it according to a standard? If yes, which standard, ... - more information would be helpful for the reader to get an understanding of how the measuremnts were designed and done.

2.3 Chemical filter analysis: What kind of analysis was done? What was the methodology? A description in more detail is needed.

3. Results and discussion:

3.1

Line 142: Sentence is hard to understand (maybe better: The aim of this study was the quantification of DD contributions for days with PM10 exceedances only)

3.2.

It is not clear for the reader, what was done according to the EC methodology and what was adapted within this study. What was the impact on the changes with respect to the results? Maybe a comparison of the results obtained by the EC methodology and the adapted EC methodology would help to understand the differences.

Line 245: "Negative values of NDL are set to zero" A discussion in more detail is necessary to guide the reader through your argument (non physical...).

3.3

line 281ff: It is hard for the reader to derive, where the threshold criteria values are extracted from. A discussion on the derivations of these values would help to understand the following arguments .

3.4.

335: typo in title: studies

How was the CL computed (derivation of results is missing)?

Why are the measurement stations different ones as compared to the other results? Do these alternative stations fulfil your criteria with respect to background (see e.g. Masenberg line 205), ...

4. Chapter 4 seems to be missing!

Author Response

The authors thank the reviewer for his appreciated recommendations to our manuscript. We tried to follow them throughout the manuscript and hope for a positive decision on the submission status.

Round  2

Reviewer 2 Report

Thanks for taking into account the comments.